**Data Availability Statement:** No datasets were generated or analysed during the current study. All

# Duration and dosing of systemic corticosteroids for acute exacerbation of COPD, protocol for a systematic review with meta-analysis of randomized trials and cohort studies

**Raymond Yin[1], Yiyang Wang[2], Yue Ying[1], Mutian Ding[1], Yunqing Ouyang[3], Emily Yuan[3], Daniel Ye[1], Shirley Yuan[1], Guanying Li[1], Winston Hou[4]***

1 Faculty of Science, The University of Western Ontario, London, Ontario, Canada, 2 University of California, Los Angeles, California, United States of America, 3 Faculty of Science, University of Toronto, Toronto, Ontario, Canada, 4 Faculty of Health Sciences, McMaster University, Hamilton, Ontario, Canada

* houwenteng@gmail.com

## Abstract

### Purpose

Acute exacerbation of chronic obstructive pulmonary disease (AECOPD) is a leading cause of deterioration in patients with otherwise stably controlled COPD. Treatments of AECOPD often require the use of corticosteroid therapy in conjunction with bronchodilators and antibiotics. However, the duration and dosage of corticosteroids still remain unclear. We propose to perform this systematic review and meta-analysis of all available randomized control trials (RCTs) and observational cohort studies to comprehensively assess the efficacy and safety of different corticosteroid duration and dosing regimen in the current body of evidence.

### Methods

We will search MEDLINE, EMBASE, CENTRAL via Ovid as well as CINAHL and Web of Science for available literature comparing different corticosteroid duration and dosage in the treatment of AECOPD. We will perform title and full text screening in duplicate, then extract relevant data using a pre-piloted extraction form. We will define short duration as less than 14-day duration of treatment and long duration as greater than 14-day treatment. We will report mortality difference as our primary outcome, with additional comparisons in incidence of re-exacerbation, hospital length of stay, lung function, incidence of hyperglycemia and infection. We will perform risk of bias assessment using the ROB2.0 and ROBINS-I tool, as well as the GRADE assessment to assess the quality of evidence.

### Results

We will publish the full results of our systematic review and meta-analysis in a peer-reviewed journal.

relevant data from this study will be made available upon study completion.

**Funding:** The author(s) received no specific funding for this work.

**Competing interests:** The authors have declared that no competing interests exist.

## Discussions

To our knowledge, this represents an updated and most comprehensive review of the literature comparing different duration and dosing regimen of corticosteroid treatments in AECOPD, as we will include both RCTs and observational studies without date or language restrictions. We aim to validate prior meta-analyses and study findings on the efficacy of short duration corticosteroid therapy over longer treatments and to inform future research directions in dosing regimens.

## Introduction

Chronic Obstructive Pulmonary Disease (COPD) is a group of conditions characterized by inflammatory changes to the lung tissue that results in persistent airway limitations [1]. These conditions include chronic bronchitis, which is mainly caused by hyperplasia of goblet cells in the upper airway that leads to hypersecretion of mucus, and emphysema, which is the parenchymal destruction of lung alveoli [2]. Currently, COPD is one of the leading causes of mortality and morbidity worldwide, with individuals older than 40, of male sex, or residing in low to middle income country at a higher risk of acquiring COPD [3, 4]. According to the Global Strategy for Diagnosis, Management, and Prevention of COPD (GOLD), COPD should be considered in any patient with a history of dyspnea, chronic cough, or sputum production. Any chronic exposure to major risk factors such as tobacco smoke or other irritating gases should warrant investigation as well [5]. Clinically, COPD is diagnosed via spirometry, with a post-bronchodilator forced expiratory volume (FEV) or forced vital capacity (FVC) of less than 0.7 indicating significant airflow limitations as outlined by GOLD [5]. Because the disease is likely irreversible, current therapies center on maintaining a stable condition, using a combination of bronchodilators, beta-2 agonists, muscarinic antagonists, and inhaled corticosteroids, depending on their disease severity [6].

Exacerbations of COPD represents a major challenge in helping patients maintain stable condition, as it may lead to increased hospital admission, worsening of their disease state, or death [5–7]. The pathophysiology underlying exacerbations of COPD is complex. Bacterial and viral respiratory infections are the main contributors to exacerbations, but environmental factors such as exposure to pollutants or airway irritants (PM2.5) are also associated with exacerbation [5, 7]. The current guideline and categorization for COPD exacerbation relies on three clinical findings of dyspnea, sputum volume and sputum purulence [5]. Mild exacerbation diagnosis is based on 1 of the above findings in addition to cough, wheezing, or respiratory infection, and short acting bronchodilator (SABD) alone should be sufficient as treatment. Moderate exacerbation is based on 2 of the above findings and is treated via SABD along with antibiotics or corticosteroids. Severe exacerbation diagnosis requires all 3 clinical findings and require complex management within a hospital or emergency room setting [5].

The role of corticosteroids has long been the point of change and contention in the management of COPD and its exacerbations [8]. Systemic steroids have been shown to reduce airway inflammations and pulmonary edema during exacerbations, thereby alleviating the clinical symptoms for patients. However, corticosteroids can also cause many significant side effects, such as fluid retention, osteoporosis, hyperglycemia, and increased risk for opportunistic infections [5, 8, 9]. The extent of both therapeutic and adverse effects of corticosteroid treatment is directly associated with the duration and dose of its administration. Current guidelines recommend a shorter, 5 to 7 days of corticosteroid administration compared to a longer, 14 days administration, based on a previous Cochrane systematic review and meta-analyses [5, 8]. However, only RCTs were

included in the analyses and the study authors have also pointed out that more data and validation is required. Additionally, guidelines and society recommendations are still unclear on the optimal corticosteroid dosage for COPD exacerbations [5, 10]. Therefore, we propose to conduct a systematic review with updated data in randomized trials as well as including cohort studies to further assess the effect of short vs long duration of corticosteroids in the treatment of COPD exacerbations. Additionally, we will also assess the effect of low vs high corticosteroid dosage on the therapeutic and adverse effects of COPD exacerbation treatment.

## Materials and methods

This systematic review and meta-analysis will be conducted in accordance with the Preferred Reporting Items for Systematic Reviews and Meta-Analyses (PRISMA) guidelines [11]. This protocol is registered within the International Prospective Register of Systematic Reviews (PROSPERO), registration ID: CRD42023374410. Any major changes to this current protocol will be reported within the review itself.

### Eligibility criteria

**Type of studies.**   We will include parallel group randomized control trials (RCTs) as well as prospective and retrospective cohort studies.

**Type of participants.**   We will include all adult patients (18 years or older) who were diagnosed with COPD exacerbation, receiving corticosteroid administration as an acute phase treatment. COPD exacerbation definition is based on individual studies. Maintenance therapy studies for stable COPD will not be included. Patients with comorbidities such as hypertension, diabetes, or other conditions will also be included.

**Type of intervention.**   We will include any corticosteroid treatment given to alleviate COPD exacerbation, via any administration method including but not limited to oral, intravenous, or mixed administration. We define short course of corticosteroid administration to be $\leq$ 7 days and long course to be >7 days based on previous studies and meta-analyses [8, 9]. In our exploratory searches on dosage studies, there have been significant heterogeneity between studies in their definition of low vs high corticosteroid dosage. We choose to define our low cumulative corticosteroid dosage at $\leq$ 300mg and high cumulative dosage as > 300mg over the entire treatment regimen. This is based on our preliminary readings of previous studies such as by Leuppi et al in 2013 [12] We will also include all corticosteroid types such as methylprednisone, prednisone, or dexamethasone. We will also include studies with any co-interventions such as antibiotics or bronchodilators, but their effects will only be explored if there are sufficient studies for analysis or if they contribute to significant heterogeneity. We will not include any placebo groups such as withholding corticosteroid treatments.

### Outcomes

We will primarily compare mortality outcomes between short versus long duration, low versus high cumulative dosage, and low versus high daily average dosage of corticosteroid treatment for COPD exacerbation. Additional outcomes will include incidence of relapse or re-exacerbations, hospital length of stay, lung function measured in $FEV_1$ or FVC, incidence of hyperglycemia and infection during or post-treatment.

### Search methods

We will conduct a systematic search of MEDLINE, EMBASE and CENTRAL via Ovid as well as CINAHL and Web of Science from inception to September 2022. We will utilize Medical

Subject Headings (MeSH) terms for broad inclusion of studies. Please refer to **S1 File** for a sample Ovid MEDLINE search strategy. We will not place any language or date restrictions in our searches and will consult colleagues for relevant translations if such needs arise. We will also perform a hand search of the reference pages in previous systematic reviews for relevant articles.

## Data collection and management

**Study screening.**   Two separate authors will perform title and abstract screening individually and in duplicate using Covidence, a web-based systematic review tool [13]. A study will only move on to full text screening if both reviewers are in agreement. Any conflicts will be resolved through discussion involving a third independent reviewer. Full text screening will also be conducted independently and in duplicate, with conflicts being resolved in the same fashion as title and abstract screening.

**Data collection.**   All data will be collected using data collection sheets developed *a priori*, with two reviewers working independently and in duplicate. Any conflicts will be resolved through discussion and third reviewer arbitration.

**Risk of bias assessment.**   The risk of bias (RoB) assessment of each included study will be done by two authors independently and in duplicate. For RCTs, RoB assessment will be conducted using RoB 2.0, a revised tool to assess risk of bias in randomized trials [14]. Two reviewers will assess bias across the five outlined domains: bias arising from randomisation process, bias due to deviations from intended interventions, bias due to missing outcome data, bias in measurement of outcome, bias in selection of the reported result. The overall RoB will be scored using the algorithm provided in the tool. For cohort studies, we will assess RoB using ROBINS-I, a tool for assessing risk of bias in nonrandomized trials for interventions [15]. The seven domains assessed will include: bias due to confounding, bias in selecting patients into the study, bias in classification of intervention, bias due to deviations from intended intervention, bias due to missing data, bias in measurement of outcomes, bias in selection of reported results. The overall risk of bias will be scored according to the algorithm provided.

**Data items.**   Below is a list of data items that are included in the pre-piloted data extraction forms.

Bibliometric data: authors, year of registration, trial registration number, digital object identifier.

Methodology: number of participating centers, location of the study, blinding methods, study duration, randomization methods.

Baseline data: total sample size, prior treatments, mean age, sex, use of adjuvant antibiotic and bronchodilator, number on mechanical ventilation before randomization, type and name of corticosteroid used, daily and cumulative dosage, follow up duration, comorbidities (i.e. hypertension, diabetes, heart failure, etc).

Outcomes: mortality (number and percentage), ICU and hospital length of stay, number of re-exacerbation during follow-up, time to re-exacerbation, FEV, FVC, number of hyperglycemic patients.

Others: other adverse events and the scales used to measure these events.

## Statistical analysis

We will first provide a narrative synthesis of the included studies, summarizing study characteristics and results both in text and table format.

We will conduct our statistical analyses using the computer program Review Manager v5.4 [16]. We will use a fixed effect model, and when the heterogeneity cannot be explained, we will perform a sensitivity analysis using a random effects model. For dichotomous outcomes such

as mortality, Mantel Haenszel odds ratios with 95% confidence intervals (CI) will be used. If the events are rare, a Peto odds ratio with 95% CI will be used. For continuous variables, mean difference (MD) with 95% CI will be used. If there are not enough meaningful data to be meta-analyzed for a particular outcome, we will qualitatively describe its results.

Heterogeneity of the included studies will be assess using a combination of visual inspection of the forest plots along with the $I^2$ statistic according to the Cochrane handbook. We will consider an $I^2 > 50\%$ to be seriously heterogenous and $>75\%$ to be very seriously heterogenous.

In the case of missing data, we will attempt to contact authors of the original study. Missing standard deviations will be inputted using methods outlined in the Cochrane Handbook for Systematic Review of Interventions using correlation coefficients.

For publication bias, we will construct funnel plots and conduct Egger's test for any analyses with more than 10 included studies.

To ascertain the confidence in cumulative effects of each individual outcome, two reviewers will perform the Grading of Recommendations, Assessment, Development and Evaluation (GRADE) analysis, independently and in duplicate. Any discrepancies will be resolved through discussion.

## Ethics and dissemination

Ethic approval is not applicable to our study since no original data will be collected. We will disseminate our results through peer-reviewed publications and conference presentations.

## Discussion

This will be an updated systematic review and meta-analyses of both RCTs and cohort studies to assess the efficacy and safety of short versus long duration and low versus high dosage of corticosteroid treatment for COPD exacerbations. Even though a shorter course of 5–7 days of corticosteroid administration has been established as noninferior to a longer course of treatment, these decisions were based on limited data and meta-analyses that included RCTs only [8, 9]. Additionally, the optimal dosing regimen for corticosteroid treatment still remains unclear. Our study hopes to expand on previous meta-analyses and incorporate more studies to assess the question of corticosteroid duration in COPD exacerbations. Further, we aim to review the current available evidence on the dosing regimen of corticosteroids and to inform future trials to rigorously assess this component of COPD exacerbation treatment.

Our review will have several strengths. First, we will not have any date or language restrictions in order to have a broad inclusion of all available studies on the subject. We will also include cohort studies in addition to randomized trials to include more patients in our meta-analysis. Additionally, we will also be assessing the efficacy and safety of short versus long corticosteroid dosing regimens, which is intrinsically linked to administration duration but less clearly explored in prior studies.

There will be several weaknesses to our review as well. The quality of our review and meta-analysis may be limited by the quality of our included studies, especially by the included cohort studies. We will rigorously assess study quality using RoB 2.0 and ROBIN-I according to the Cochrane Handbook. Additionally, because corticosteroid dosing is less clearly defined and may vary between studies, we expect heterogeneity within our included studies.

Despite these limitations, we are confident that our systematic review and meta-analysis will be the most comprehensive quantitative and qualitative synthesis of available evidence on both the duration and dosing of corticosteroid treatment of COPD exacerbations. Our study hope to help build further confidence in physicians treating COPD exacerbations and may inform future research directions in terms of corticosteroid dosing regimens.

## Supporting information

**S1 Checklist. PRISMA-P 2015 checklist.**
(DOCX)

**S1 File. MEDLINE search strategy example.**
(PDF)

## Acknowledgments

We wish to offer our special thanks to Dr. Weiting Xiong, MD, for dedicating his time to review and provide guidance on the protocol.

## Author Contributions

**Conceptualization:** Raymond Yin, Yiyang Wang, Winston Hou.

**Data curation:** Raymond Yin, Yiyang Wang, Yue Ying, Mutian Ding, Shirley Yuan, Guanying Li, Winston Hou.

**Investigation:** Raymond Yin, Yiyang Wang, Yue Ying, Mutian Ding, Yunqing Ouyang, Emily Yuan, Daniel Ye.

**Methodology:** Yiyang Wang, Yue Ying, Emily Yuan, Winston Hou.

**Project administration:** Winston Hou.

**Validation:** Raymond Yin, Yiyang Wang, Mutian Ding, Winston Hou.

**Writing – original draft:** Raymond Yin, Yiyang Wang, Yue Ying, Mutian Ding, Yunqing Ouyang, Emily Yuan, Winston Hou.

**Writing – review & editing:** Raymond Yin, Yue Ying, Daniel Ye, Shirley Yuan, Guanying Li, Winston Hou.

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
