## [Decision Letter · Decision Letter 0]

29 Jun 2023

PONE-D-23-05685Duration and dosing of systemic corticosteroids for acute exacerbation of COPD, protocol for a systematic review with meta-analysis of randomized trials and cohort studies.PLOS ONE

Dear Dr. Hou,

Thank you for submitting your manuscript to PLOS ONE. After careful consideration, we feel that it has merit but does not fully meet PLOS ONE’s publication criteria as it currently stands. Therefore, we invite you to submit a revised version of the manuscript that addresses the points raised during the review process.

We look forward to receiving your revised manuscript.

Kind regards,

Jung Yeon Heo

Academic Editor

PLOS ONE

Journal Requirements:

Additional Editor Comments:

This is a study protocol for systematic review and potential meta-analysis on duration and dosing of systemic corticosteroid in COPD patients with acute exacerbation. The manuscript is well written, and study background and analysis methodology are clearly presented. Overall, the area is of interest and might be relevant to medical specialists like pulmonologist. I have a few minor comments that the authors might consider.

Given study background, study key points will be to suggest which duration and dose are optimal between short and long-course therapy, and between high-dose and low-dose therapy in COPD patients with acute exacerbation. However, high and low-dose steroid therapy was not predefined because of heterogeneous findings in preliminary searches. In addition, short course therapy was defined as administration to be ≤ 14 days whereas long course was > 14 days. However, steroid duration of 2-5 day regimen was compared to 14 days regimen in the cited reference. Generally, the optimal duration of systemic glucocorticoid therapy is known to be 5 to 14 days. Thus, it would be better to redefine steroid duration and dose with COPD experts.

Please note that I have acted as a reviewer for this manuscript, and you will find my comments below, under Reviewer 3.

Reviewers' comments:

Reviewer's Responses to Questions

**Comments to the Author**

1. Does the manuscript provide a valid rationale for the proposed study, with clearly identified and justified research questions?

Reviewer #1: Yes

Reviewer #2: Yes

Reviewer #3: Yes

2. Is the protocol technically sound and planned in a manner that will lead to a meaningful outcome and allow testing the stated hypotheses?

Reviewer #1: Yes

Reviewer #2: Yes

Reviewer #3: Yes

3. Is the methodology feasible and described in sufficient detail to allow the work to be replicable?

Reviewer #1: Yes

Reviewer #2: Yes

Reviewer #3: Yes

4. Have the authors described where all data underlying the findings will be made available when the study is complete?

Reviewer #1: Yes

Reviewer #2: Yes

Reviewer #3: Yes

5. Is the manuscript presented in an intelligible fashion and written in standard English?

Reviewer #1: Yes

Reviewer #2: Yes

Reviewer #3: Yes

6. Review Comments to the Author

You may also provide optional suggestions and comments to authors that they might find helpful in planning their study.

Reviewer #1: This protocol looks well-organized for Systematic Review and meta-analysis aligned with PROSPERO registry. All the requirements are accurately stated with the PRISMA checklist. Look forward to see the finalized result in an well-designed article.

Reviewer #2: In your Methods description, comorbidities are not mentioned: they are quite relevant in evaluating effects and side effects od CS in COPD. Some words should be added

Reviewer #3: This protocol can be useful to conduct the meta-analysis for steroid use in the patients with acute exacerbation of COPD.

7. PLOS authors have the option to publish the peer review history of their article (what does this mean?). If published, this will include your full peer review and any attached files.

Reviewer #1: No

Reviewer #2: No

Reviewer #3: No

---

## [Author Response · Author response to Decision Letter 0]

1 Aug 2023

Dear editor and reviewers,

I have attached the newly revised version of our manuscript titled “Duration and dosing of systemic corticosteroids for acute exacerbation of COPD, protocol for a systematic review with meta-analysis of randomized trials and cohort studies” for publication at PLOS One. You will also find in the same submission a marked-up copy of the manuscript as well as an unmarked version of the revised paper. 

We would like to give thanks for both the editor and the reviewers for their insightful comment. We have addressed them in our manuscript. The detailed changes are outlined in the paragraphs below as well as in the marked-up version of the manuscript. We would like to mention that we were not able to provide the minimal data set underlying the results, as this is a protocol paper and no minimal data set exist. 

We would like to thank reviewer #1’s comment, and for taking their time in looking over the manuscript. 

We have addressed the issue raised by reviewer #2 in our manuscript. We have added sentences in the materials and methods section (type of participants and data item sub-section) on extracting data regarding the comorbidities of these patients as well as the side effects, such as hypertension, diabetes, and heart failure. Additionally, we have added comments on extracting side effects post-corticosteroid administration. These comorbidities are common among COPD patients, and we thank the reviewer for pointing that out. 

We have also addressed reviewer #3 / the editor’s comments on the protocol. After further consultation with an internal medicine specialist, we have now clearly defined the dosage cut-offs as well as refined our definition for short vs long duration corticosteroid administration. Short duration is now defined as ≤ 7 days whereas long duration is > 7 days. For cumulative dosage, low is defined as ≤ 300mg total and high is defined as > 300mg total, based on our preliminary readings of prior studies. These will be reflected in our final analysis, and we give our thanks to the editor. 

Sincerely and thank you. 

Winston Hou, corresponding author.

---

## [Editor Report · Decision Letter 1]

21 Aug 2023

Duration and dosing of systemic corticosteroids for acute exacerbation of COPD, protocol for a systematic review with meta-analysis of randomized trials and cohort studies.

PONE-D-23-05685R1

Dear Dr. Hou,

We’re pleased to inform you that your manuscript has been judged scientifically suitable for publication and will be formally accepted for publication once it meets all outstanding technical requirements.

Kind regards,

Jung Yeon Heo

Academic Editor

PLOS ONE

Additional Editor Comments (optional):

Thank you for submitting a protocol revised on the basis of reviewers comments
---

## [Editor Report · Acceptance letter]

22 Aug 2023

PONE-D-23-05685R1 

Duration and dosing of systemic corticosteroids for acute exacerbation of COPD, protocol for a systematic review with meta-analysis of randomized trials and cohort studies. 

Dear Dr. Hou:

I'm pleased to inform you that your manuscript has been deemed suitable for publication in PLOS ONE. Congratulations! Your manuscript is now with our production department. 

Kind regards, 

on behalf of

Dr. Jung Yeon Heo 

Academic Editor

PLOS ONE